# Role of Maternal Diet in the Risk of Childhood Acute Leukemia: A Systematic Review and Meta-Analysis

**DOI:** 10.3390/ijerph20075428

**Published:** 2023-04-06

**Authors:** Jessica Blanco-Lopez, Isabel Iguacel, Silvia Pisanu, Claudia Choma Bettega Almeida, Eva Steliarova-Foucher, Ciska Sierens, Marc J. Gunter, Elena J. Ladas, Ronald D. Barr, Koen Van Herck, Inge Huybrechts

**Affiliations:** 1International Agency for Research on Cancer, 69007 Lyon, France; 2Faculty of Health Sciences, University of Zaragoza, 50009 Zaragoza, Spain; 3Department of Biomedical Sciences, Section of Microbiology and Virology, University of Cagliari, 09124 Cagliari, Italy; 4Department of Nutrition, Federal University of Paraná, Curitiba 1299, Brazil; 5Department of Public Health and Primary Care, Ghent University, 9000 Ghent, Belgium; 6Faculty of Medicine, School of Public Health, Imperial College London, London SW7 2AZ, UK; 7Division of Hematology, Oncology, and Stem Cell Transplantation, Department of Pediatrics, Columbia University Irving Medical Center, New York, NY 10032, USA; 8Departments of Pediatrics, Pathology and Medicine, McMaster University, Hamilton, ON L8S 4L7, Canada

**Keywords:** acute lymphoblastic leukemia, acute myeloid leukemia, maternal diet, prevention, risk factor

## Abstract

Many studies have investigated the etiology of acute leukemia, one of the most common types of cancer in children; however, there is a lack of clarity regarding preventable risk factors. This systematic review and meta-analysis aimed to summarize the current evidence regarding the role of maternal dietary factors in the development of childhood leukemia. All epidemiological studies published until July 2022 that evaluated maternal dietary risk factors for childhood acute leukemia were identified in two electronic databases (PubMed and Web of Science) without limits of publication year or language. A total of 38 studies (1 prospective cohort study, 34 case-control studies and 3 studies with pooled analysis) were included. The published risk estimates were combined into a meta-analysis, using the Generic Inverse Variance method. The maternal consumption of fruits (two or more daily servings vs. less) was inversely associated with acute lymphoblastic leukemia (odds ratio = 0.71; 95% CI, 0.59–0.86), whereas maternal coffee intake (higher than two cups per day vs. no consumption) was associated with an increased risk of acute lymphoblastic leukemia (odds ratio = 1.45; 95% CI, 1.12–1.89). Despite these findings, more high-quality research from cohort studies and the identification of causal factors are needed to develop evidence-based and cost-effective prevention strategies applicable at the population level. Review Registration: PROSPERO registration no. CRD42019128937.

## 1. Introduction

Acute leukemia is one of the most frequently diagnosed cancers among children and adolescents. Its worldwide incidence is approximately 46 cases per million per year in children aged 0–14 years and approximately 29 cases per million in adolescents aged 15–19 years [1]. In many populations of children aged 0–14 years, almost 80% of cases have acute lymphoblastic leukemia (ALL), while 15% have acute myeloid leukemia (AML) [2].

The etiology of childhood acute leukemia is linked to X-ray radiation exposure, specific forms of chemotherapy and some genetic syndromes [3]. However, all these risk factors together may explain only 10% of cases [4]. Although acute leukemia is the most common cancer in children and adolescents, it is a rare disease; therefore, studying its etiology remains challenging. Nonetheless, a causal mechanism for the most common variant (ALL) has been proposed, which incorporates both antenatal and postnatal factors [5].

Considering the differential incidence of acute leukemia by geographic region worldwide, the potential roles of lifestyle and environmental factors in its etiology have been proposed [1]. A peak incidence occurs at approximately two-five years of age, and it is reasonable to assume that factors in the prenatal or perinatal period affect the pathogenesis of childhood leukemia [3,6]. Indeed, several studies have demonstrated an increased risk of leukemia due to maternal smoking and alcohol consumption during pregnancy [7,8]. However, little is known about the impact of other lifestyle factors such as the mother’s overall diet during pregnancy on the risk of childhood leukemia. 

The current systematic review and meta-analysis focused on the role of maternal diet in the risk of childhood acute leukemia. It provides an overview of existing data linking the influence of maternal diet, supplementation and the consumption of non-alcoholic beverages to the development of acute leukemia in children. We used combined evidence to assess the potential impact of maternal diet on the risk of childhood acute leukemia. 

## 2. Materials and Methods

### 2.1. Definition of the Outcome 

The main outcome of our systematic review is acute leukemia (AL) incidence in children aged 0–14 years and adolescents aged 15–19 years. The leukemia types considered included acute lymphoblastic leukemia (ALL) and acute non-lymphoblastic leukemia (ANLL), mainly acute myeloid leukemia (AML). 

### 2.2. Definition of the Risk Factors 

The exposures included maternal dietary and non-alcoholic drink intake and supplements consumed during the peri-pregnancy period.

### 2.3. Systematic Review Registration 

Details of the protocol for this systematic review were registered on PROSPERO, registration number CRD42019128937. 

### 2.4. Search Strategy 

To identify the maternal dietary factors that have been studied in relation to childhood leukemia risk, we performed a multi-tiered electronic search of PubMed and Web of Science. First, an explorative search for studies on leukemia in infants, children and adolescents was conducted in December 2017; this was not restricted by language or publication date. A total of 66 studies were identified in this search. Based on the results of this first search, we were able to narrow down the keywords further and perform a more targeted search in July 2022, which focused on some identified maternal exposures such as dietary intake (food groups and micronutrients), consumption of non-alcoholic beverages (coffee, tea and soft drinks) and supplements used during pregnancy, and no limit of publication year or language restriction was used. For all constructs, database-specific thesauri were used to identify the relevant synonyms. The search strategy is illustrated in Figure 1.

### 2.5. Selection of the Studies and Quality Assessment

Studies were eligible for inclusion if they reported effect estimates (relative risk (RR), odds ratio (OR) or hazard ratio (HR) and 95% confidence intervals (CI)) or if they provided sufficient data to calculate the estimates for different levels of exposure. 

Table 1 lists the inclusion and exclusion criteria used in the review. The PRISMA guidelines were followed [9] (Appendix A). All references found through the search questions (Figure 1) were imported into the Endnote software program (Version x8, Clarivate Analytics, Philadelphia, PA, USA). Two authors independently searched each database (Pubmed and Web of Science) for publications. Any uncertainty regarding the selection of a given article was resolved after screening by a third reviewer (IH) and discussed in the working group, if needed. The final eligibility criteria for the included studies are listed according to the PICOS (Table 2), and the selected studies included for each extracted factor are summarized in Table 3.

Subsequently, quality control of the selected articles was performed using the checklist by Fowkes et al. to ensure the inclusion of high-quality studies [10]. This checklist includes questions on the study design and sample, control group characteristics, quality of measurements and outcomes, completeness and influences of distortions. A major flaw identified for a given criterion is flagged by the sign ++. The + sign was applied to a criterion with a minor flaw. In case of no flaws, nil (0) was applied. If a study scored ++ on more than one criterion of this checklist, it was excluded (Figure 1). The quality control results of the selected articles are shown in Appendix A.

### 2.6. Data Extraction

Table 4 summarizes the included studies by name, design, sample size, age at diagnosis and country of study population. The extracted key data from all included studies, effect estimates (RR, OR, or HR, 95% CI) and tested confounding variables are presented in Table 5, Table 6, Table 7, Table 8, Table 9 and Table 10. The results from the pooled studies were also included, as reported in the summary tables.

### 2.7. Quantitative Meta-Analysis

A quantitative analysis of the risk of AL, ALL, or AML in children was performed to determine the risk factors that met the eligibility criteria for the meta-analysis. For each included factor, the choice of performing an additional quantitative analysis was based on the number (at least two studies) and homogeneity of the included studies (e.g., results from studies restricted to infants or from univariate analysis with no adjustments were not pooled with the rest of the results) and overlapping study data were also excluded. We used the adjusted odds ratio (OR) reported in case-control studies and the relative risk (RR) reported in cohort studies to calculate the overall effect. The analyses were conducted using the Generic Inverse Variance method, in which the weight of each study was equivalent to the inverse of the variance of the effect estimate [47]. A random-effects model was used if there was evidence of heterogeneity. Some exposures were reported at different levels; consequently, we calculated an estimate for binary exposure status (exposed vs. not exposed) by pooling exposed cases and controls across all levels of exposure. I2 statistics were used to assess heterogeneity, where 0% indicated perfect homogeneity and 100% complete heterogeneity [48]. Data were analyzed using Review Manager (RevMan) V.5.4.1, Cochrane, London, UK [49]. The results are summarized in Table 11.

## 3. Results

### 3.1. Selected Studies

The results of the search strategy and process of selecting studies for review are illustrated in Figure 1. The preliminary search returned 7813 studies, which were screened based on the title. Following the assessment of titles and abstracts, 317 articles met the criteria for full-text review. The subsequent screening of the full reports of the 317 selected publications resulted in the selection of 38 eligible studies.

### 3.2. Study Characteristics

Table 3 provides an overview of the number of studies included in each extracted exposure. The detailed characteristics (authorship, year of publication, study design, period and place of the study, number and age of participants, control source and extracted exposure variables) of each included study are shown in Table 4.

### 3.3. Quality Assessment

The quality assessment [10] for the included studies is shown in Appendix A. The most common weakness of many studies was the lack of adjustment for possible confounding factors such as maternal smoking status, maternal alcohol consumption and perinatal factors. The confounding factors used in the analysis for each included study are shown in Table 5, Table 6, Table 7, Table 8, Table 9 and Table 10.

### 3.4. Maternal Intake

#### 3.4.1. Food Group Intakes

Seven case-control studies [11,12,13,14,15,16,17] investigated the association between maternal dietary intake and the risk of AL in children. The results are summarized in Table 5.

Two studies reported results for AL, one focused on cured meats but their results did not achieve statistical significance [16] and the other reported a statistically significant increased risk of leukemia in infants with the maternal daily consumption of fresh vegetables identified a priori as containing DNA topoisomerase II inhibitors [15].

Regarding ALL, three consecutive studies [11,14,17] conducted by the Northern California Childhood Leukemia Study (NCCLS) reported a statistically significant inverse association with a higher daily consumption of vegetables, fruits and proteins. Further analysis determined that these associations were statistically significant for carrots, cantaloupe, oranges, green beans, beans and beef. A case-control study [13] reported statistically significant inverse associations between the consumption of fruits, vegetables and fish during pregnancy and the risk of ALL in children. Conversely, a higher intake of sugar and syrup and meat and meat products had a statistically significantly increased risk of ALL. Two other studies [12,15] did not report statistically significant associations between the consumption of cured meats and ALL. A meta-analysis of two studies [13,17] was performed, in which a statistically significant inverse estimate for ALL was found for higher maternal consumption of fruits (OR = 0.71, 95% CI: 0.59–0.86), but not for vegetables (OR = 0.88, 95% CI: 0.69–1.11), and of two [12,14] studies for cured meats, which was not significant (OR = 1.01, 95% CI: 0.68–1.51) (Table 11).

A statistically significant increased risk for AML was reported in one case-control study related to the intake of certain fresh vegetables (e.g., vegetables such as onions, broccoli and asparagus that are rich in Quercetin, which induces human DNA topoisomerase II) [15], and another study reported an inverse association with fruit consumption [17].

#### 3.4.2. Non-Alcoholic Drinks

A.Coffee

Two pooled studies [18,19], eight case-control studies [15,20,21,22,23,24,25,26] and one cohort study [27] evaluated the association between the maternal consumption of coffee during pregnancy and the risk of AL (Table 6).

A cohort study [27] (N = 141,216/AL cases = 96) reported no statistically significant association between coffee intake during pregnancy and the risk of AL. In the present meta-analysis, including five studies [22,24,25,26,27] that compared the highest vs. the lowest consumption reported for coffee intake during pregnancy, we did not observe a statistically significant association for AL (OR = 1.36, 95% CI: 0.97–1.91).

In a pooled study [19] of eight case-control studies (from published [20,21,22,23,24] and unpublished data), an increased risk of ALL in children whose mother’s coffee intake was more than two cups per day in comparison with no consumption (OR = 1.27, 95% CI: 1.09–1.43, *p* = 0.005) was reported. A meta-analysis of five studies [20,21,22,23,24] comparing the highest vs. the lowest consumption reported a similar result (OR = 1.45, 95% CI: 1.12–1.89).

Another pooled study [18] of eight case-control studies reported an increased risk of AML for children whose mothers consumed more than one cup per day (OR = 1.40, 95% CI: 1.03–1.92), and the risk increased with increments of one cup per day (OR = 1.18, 95% CI: 1.01–1.39). A meta-analysis of four studies comparing the highest versus the lowest consumption for AML [20,22,23,24] reported that the association was not statistically significant (OR = 1.18, 95% CI: 0.66–2.13) (Table 11, Figure 2a–c).

B.Tea

Two pooled analyses [18,19] and five case-control studies [15,20,21,22,23,24] evaluated the association between maternal tea consumption during pregnancy and the risk of AL (Table 7).

Overall, the case control studies [15,20,21,22,23,24] did not report any statistically significant associations between maternal tea intake during pregnancy and the risk of childhood AL. Both pooled analyses looked for dose responses and interactions, but the findings were null [18,19]. A recent cohort study also added tea intake to their model and found no association (data not reported) [27]. A meta-analysis of two [22,24] studies found no statistically significant association estimate for AL with the highest vs. the lowest consumption of tea (OR = 0.97, 95% CI: 0.80–1.51) (Table 11).

C.Soft drinks

Three case-control studies [15,16,22] evaluated the association between maternal intake of soft drinks (mostly cola beverages) and AL risk (Table 8). Only one case-control study [22] reported an increased risk with the maternal intake of cola beverages and AL, especially for AML. A meta-analysis of the three studies found marginal association estimates for AL with the highest vs. the lowest consumption of cola beverages (OR = 1.23, 95% CI: 0.97–1.55) (Table 11).

### 3.5. Nutrients

#### 3.5.1. Micronutrient Intake

Four case-control studies [11,14,28,29] evaluated the association between maternal micronutrient intake and the risk of childhood AL (Table 9). Two of these studies from the NCCLS reported estimations with micronutrients from food only. One [11] found that the total glutathione level was inversely associated with ALL. Another study reported an inverse association between the daily vitamin B6 intake dose and the risk of AML [29].

#### 3.5.2. Supplements of Vitamins and Minerals

Twenty case-control studies [11,14,28,29,30,31,32,33,34,35,36,37,38,39,40,41,42,43,44,45] and one pooled study [46] evaluated the association between vitamin and mineral supplementation during pregnancy and the risk of AL in children. The results are compared in Table 10.

Folic acid supplementation and the risk of AL have been widely studied [11,14,28,29,31,40,41,42,43,45]. A pooled study [46], which included published [11,14,41,42,43,45] and unpublished data, reported an inverse association between maternal folic acid for ALL (OR = 0.77, 95% CI: 0.67–0.88) and AML (OR = 0.52, 95% CI: 0.31–0.89) with supplementation during pregnancy versus no supplementation. A meta-analysis performed on two studies [40,41] (OR = 0.55, 95% CI: 0.28–1.09) for AL and on six studies [28,29,40,41,43,45] (OR = 0.77, 95% CI: 0.59–1.01) for ALL did not yield statistically significant results (Table 11, Figure 3a–c).

Supplementation with iron during pregnancy and the risk of childhood leukemia in offspring have also been studied [11,14,30,32,34,35,36,37,39,43,45]. Most studies have reported non-statistically significant associations with AL, ALL, or AML. A meta-analysis of iron supplementation during pregnancy in three studies [32,34,35] for AL (OR = 0.94, 95% CI: 0.65–1.38) and seven studies [30,32,35,39,42,43,45] for ALL (OR = 0.96, 95% CI: 0.76–1.21) found no statistically significant results (Table 11, Figure 4a,b).

Supplementation with multivitamins during pregnancy has been reported [33,36,37,38,42,43,44]. A pooled analysis [46] that included published [37,38,42,43] and unpublished data reported an inverse association for ALL (OR = 0.81, 95% CI: 0.74–0.88) with supplementation during pregnancy. A meta-analysis was not performed for this supplementation owing to the heterogeneity of the data.

## 4. Discussion

### 4.1. Summary of the Main Findings

The main findings of the systematic review suggest that maternal dietary intake rich in vegetables, fruits and protein sources (beans and beef) and supplementation with folic acid and multivitamins during pregnancy could have a protective effect against childhood acute leukemia. The increased intake of cured meats, coffee and/or caffeinated beverages could increase the risk of childhood acute leukemia. Some of these findings were confirmed by our meta-analysis; a higher maternal consumption of fruits was significantly inversely associated with ALL (OR = 0.71, 95% CI: 0.59–0.86). This association was recently found in a previous meta-analysis [50] in which fruit (OR = 0.81, 95% CI: 0.67–0.99), vegetables (OR: 0.51, 95% CI: 0.28,0.94) and legumes (OR = 0.76, 95% CI: 0.62–0.94) were inversely associated with ALL.

Our meta-analysis also confirmed that regular maternal coffee intake higher than two cups per day was associated with an increased risk of ALL (OR = 1.45; 95% CI, 1.12–1.89). Similarly, in a meta-analysis published in 2014 [51] maternal coffee consumption (high drinkers vs. non/lowest drinkers) was associated with an increased risk of ALL (OR = 1.65, 95% CI: 1.28–2.12) and childhood AML (OR = 1.58, 95% CI: 1.20–2.08). The results of other meta-analyses published in 2015 [52] on maternal coffee consumption (high drinkers vs. non/lowest drinkers) were associated with an increased risk of AL (OR = 1.57, 95% CI: 1.14–2.11) and ALL (OR = 1.43, 95% CI: 1.22–1.68) but not AML (OR = 1.81, 95% CI: 0.93–3.53). The results of a meta-analysis published in 2016 [53] found an increased risk of ALL associated with coffee drinking during pregnancy (adjusted OR = 1.44, 95% CI: 1.07–1.92).

Regarding maternal tea consumption during pregnancy, our analysis did not find a statistically significant association. These findings are consistent with those of a previous meta-analysis [52].

Regarding soft drink intake during pregnancy, the analysis suggested a possible increased risk for AL, but this finding was not significant. A previous meta-analysis [52] reported an increased risk between low and moderate consumption for AL (OR = 1.21, 95% CI: 1.03–1.44) and ALL (OR = 1.31, 95% CI: 1.09–1.59).

Folic acid supplementation during pregnancy suggested a modest inverse association with ALL (OR = 0.77; 95% CI: 0.59–1.01). A recent systematic review reported that the use of maternal folic acid supplementation before pregnancy was associated with a 30% decreased risk of ALL (OR = 0.69, 95% CI: 0.50–0.95) and the use of vitamin supplements before pregnancy was associated with a 20% decreased risk of ALL (OR = 0.81, 95% CI: 0.74–0.88) [50]. A more recent meta-analysis reported an inverse association between folic acid supplementation and ALL (OR = 0.75, 95% CI: 0.66–0.86) [54].

### 4.2. Potential Mechanisms

The inverse association between maternal fruit and vegetable intake and the risk of AL could be explained mainly by the high content of protective bioactive components such as micronutrients and fiber. Some fruits and vegetables that were found to be protective in some studies [11,14] are rich in carotenoids, flavonoids and vitamins A and C, which are important for DNA repair [55,56]. Also, the presence of the antioxidant tripeptide glutathione, present in meat and vegetables, plays a critical role in many cellular processes such as cell differentiation, proliferation and apoptosis [57].

Some studies suggest the association between the maternal intake of cured meats and the risk of childhood AL. This is because of the presence of N-nitroso precursors in these meats that can be converted to carcinogenic components [58] by acids in the stomach and then transported through the placenta to the developing child. This might increase the risk of malignancy as a result [59].

The decreased risk of ALL associated with the maternal use of vitamin formulations as supplements can be explained by the capacities of the components, including folate and iron, to protect against oxidative damage to lipids, lipoproteins and DNA [56]. On the other hand, folic acid and its derivatives, known as folates, are chemo-protective micronutrients with an essential role in the maintenance of health and genomic integrity [60].

The increased risk of AL associated with the maternal consumption of coffee and other caffeinated drinks could potentially be explained by the effects of their components (caffeine and chlorogenic acid) in inhibiting Topoisomerase I and Topoisomerase II [61,62], which are nuclear enzymes that solve the topological problems associated with DNA replication, transcription, recombination and chromatin remodeling [63]. Tea contains lower levels of caffeine, folate and polyphenols, which might explain the divergent results of coffee consumption.

### 4.3. Strengths and Limitations

The present systematic review and meta-analysis integrated the available published information regarding maternal dietary intake and the risk of childhood AL. Unfortunately, most of the evidence comes from case-control studies (which are prone to recall bias) that were conducted more than a decade ago and restricted to high-income countries. It is also important to note that the accurate evaluation of dietary intake is challenging. We also noticed that some studies restricted their search to specific food items [18] and, given the heterogeneity of how results were reported (above all the studies), this makes it difficult to compare results.

However, there are new generation studies (biomarker-based, which are not the focus of this review) that can support some of these findings. Studies have been conducted to assess the presence of carcinogens (such as acrylamide, nitrosamines and others) in the maternal diet, which can be found in the placenta, linked to fetal hemoglobin, and can also be found in newborn lymphocytes. Since these biomarkers were also found in healthy children, further studies focusing on genetic variants found some polymorphisms that could affect the expression of enzymes such as folate hydrolase 1. With lower detoxifying activity, DNA damage or the higher activity of carcinogens may occur [64].

Regardless of the limitations of these studies, professionals dedicated to treating childhood cancer have demonstrated a great capacity to collaborate and collect high-quality data and biospecimens. This could lead to improvements in the quantity and quality of future research, for more prospective cohort studies and a more comprehensive and accurate assessment of the role of nutrition in carcinogenesis are needed.

## 5. Conclusions

This systematic review of the available literature indicates that a maternal diet rich in vegetables, fruits and supplementation with folic acid and multivitamins during pregnancy could be inversely associated with the risk of childhood AL, while a higher intake of coffee and/or caffeinated beverages could increase the risk. Only the results regarding maternal diet rich in fruits and the consumption of coffee in relation with childhood AL were confirmed by the meta-analysis performed on the available published data. Although this review sheds light on the risk factors related to maternal diets and the prevention of cancers in children and adolescents, more studies are needed to confirm causality and the potential mechanisms involved.

## Figures and Tables

**Figure 1 ijerph-20-05428-f001:**
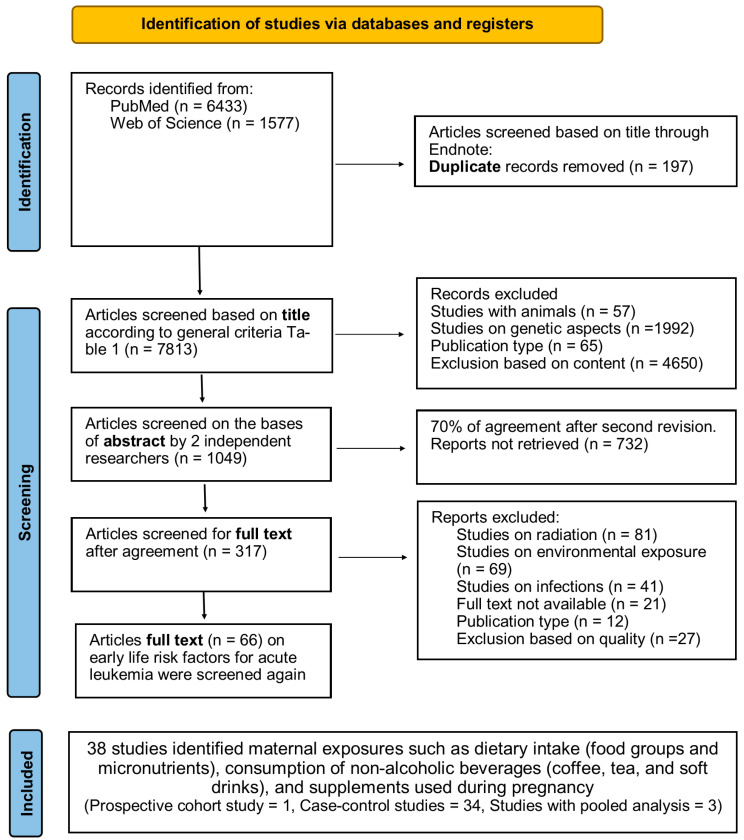
Search Strategy.

**Figure 2 ijerph-20-05428-f002:**
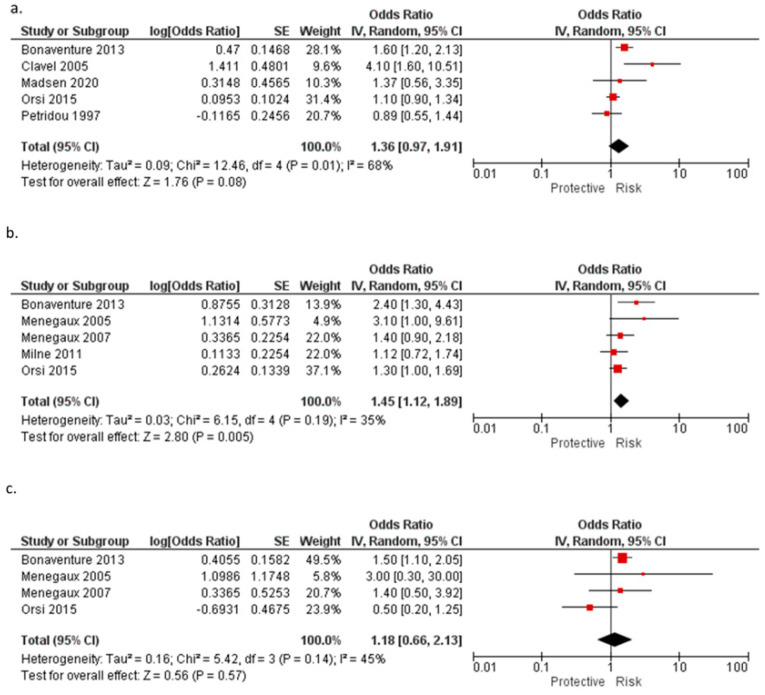
Meta-analysis of the association between the consumption of coffee during pregnancy and (**a**) acute leukemia [22,24,25,26,27], (**b**) acute lymphoblastic leukemia [20,21,22,23,24] and (**c**) acute myeloid leukemia [20,21,22,24]. Note: the individual estimate (OR) from the studies are represented by the red box, and the black diamond represents the estimate of the meta-analysis.

**Figure 3 ijerph-20-05428-f003:**
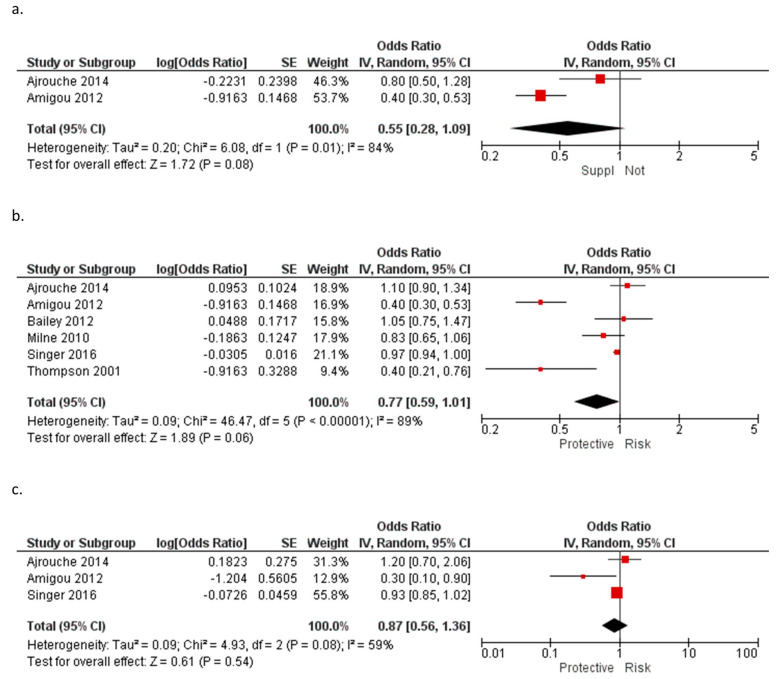
Meta-analysis of the association between the supplementation with folic acid during pregnancy and (**a**) acute leukemia [40,41], (**b**) acute lymphoblastic leukemia [28,29,40,41,43,45] and (**c**) acute myeloid leukemia [29,40,41]. Note: the individual estimate (OR) from the studies are represented by the red box, and the black diamond represent the estimate of the meta-analysis.

**Figure 4 ijerph-20-05428-f004:**
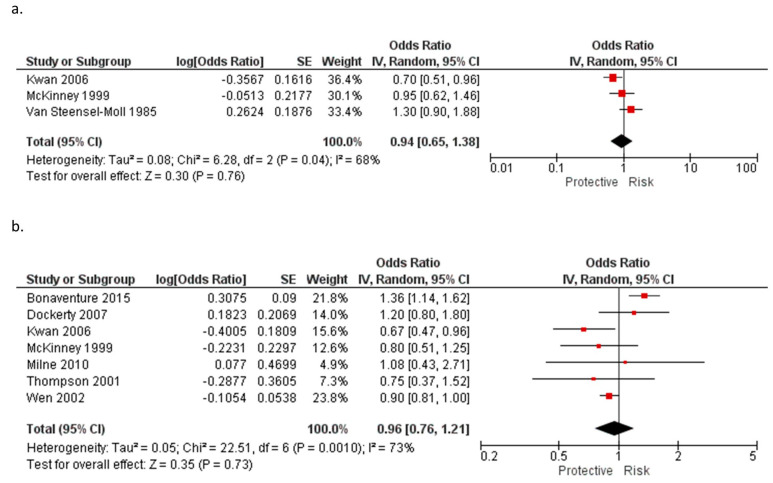
Meta-analysis of the association between the supplementation with iron during pregnancy and (**a**) acute leukemia [32,34,35] and (**b**) acute lymphoblastic leukemia [30,32,35,39,42,43,45]. Note: the individual estimate (OR) from the studies are represented by the red box, and the black diamond represent the estimate of the meta-analysis.

**Table 1 ijerph-20-05428-t001:** Inclusion and exclusion criteria.

General inclusion criteria
All studies describing the maternal dietary factors and their influence on the incidence of acute leukemia in children and adolescents were included in the systematic review. The risk factors considered were maternal nutrition (food and beverages) and supplement use during pregnancy. Pooled analysis including new data was also included in the review.
General exclusion criteria
Studies on animals Studies on the genetic aspects Full text not available Studies without original data (editorial, review, reports, guidelines)
Exclusion criteria based on content
Studies of leukemia in children who already have other concomitant disorders (e.g., Down syndrome, congenital malformations) Studies of the association between radiation and leukemia (e.g., prenatal boot setting for X-rays during pregnancy; living near a nuclear power station) Studies of the association between environmental exposure and leukemia (e.g., air pollution, exhaust fumes, pesticides) Studies on the association between infectious diseases and leukemia (maternal infections during pregnancy, illness of the child in the early years of life, presence in the nursery)
Exclusion criteria based on study scope and data quality
Small studies (e.g., fewer than 50 cases) Insufficient quality based on the checklist of Fowkes et al.

**Table 2 ijerph-20-05428-t002:** PICOS criteria for inclusion and exclusion of studies.

Parameter	Criterion
Participants	Children and adolescents diagnosed with acute leukemia
Interventions	Maternal dietary intake (including food groups and caffeinated beverages) and supplements
Control/comparator group	Healthy children and adolescents
Outcomes	Childhood acute leukemia
Study design	Observational studies with a comparison group (cohort studies, case-control studies)

**Table 3 ijerph-20-05428-t003:** Overview of the included studies.

Extracted Factors (Maternal Factors during Pregnancy)	Number of Included Articles ^1^	References
Food group intake (vegetables, fruits, cereals and grains, meats, diary)	7	[11,12,13,14,15,16,17]
Coffee intake	11	[15,18,19,20,21,22,23,24,25,26,27]
Tea intake	7	[15,18,19,21,22,23,24]
Soft drink intake (cola drinks)	3	[15,16,22]
Micronutrient intake (glutathione, carotenoids, folates, etc.)	4	[11,14,28,29]
Supplements of vitamins and minerals (folic acid, iron, etc.)	21	[11,14,28,29,30,31,32,33,34,35,36,37,38,39,40,41,42,43,44,45,46]

^1^ A total of 38 studies were included in total (1 prospective cohort study, 3 pooled analysis studies and 34 case-control studies).

**Table 4 ijerph-20-05428-t004:** Characteristics of the included studies.

Study/Reference	Location and Study Period	Leukemia Type	Age Range (Years)	Cases/Controls	Control Source	Maternal Exposure Variable(s)	Source(s)/Assessment Tool
Cohort							
Madsen et al. [27]	Denmark, 1996–2002	Acute leukemia	0–14	96/Cohort 141, 216		Coffee and tea consumption	Health registries
Case control							
Abudaowd et al. [31]	Saudi Arabia, 2008–2019	ALL, AML	1–19	74/148	Community controls matched on age and gender	Use of supplements	Structured telephone interview
Ajrouche et al. [40] (ESTELLE ^a^)	France, 2010–2011	ALL, AML	0–14	747/1421	Population controls matched on age and gender	Use of supplements	Structured telephone questionnaires
Amigou et al. [41] (ESCALE ^a^)	France, 2003–2004	ALL, AML	0–14	764/1681	Randomly selected from French households with a landline telephone	Use of supplements	Standardized telephone interview
Bailey et al. [28] (Aus-ALL ^b^)	Australia, 2003–2007	ALL	0–14	333/695	Prospectively recruited by random digit dialing and matched on age, sex and residence	Micronutrients and supplements	Mailed, self-administered questionnaires and Food Frequency Questionaries
Bonaventure et al. [22] 2013 (ESCALE ^a^)	France, 2003–2004	ALL, AML	0–14	764/1681	Randomly selected with a landline telephone	Coffee, tea and cola drink consumption	Interviewed with standardized questionnaires
Bonaventure et al. [30] 2015 (UKCCS ^c^)	UK, 1991–1996	ALL, AML	0–14	864/2524	Community controls matched on sex, month and year of birth and region of residence	Use of supplements	Primary care (general practice) records
Clavel et al. [26] (ADELE ^a^)	France, 1995–1999	ALL, AML	0–14	247/288	Hospitalized controls (for diseases other than cancer or birth defects) matched on age, gender and center	Coffee consumption	Interviewer-administered questionnaire
Dockerty et al. [42]	New Zealand, 1990–1993	ALL	0–14	97/303	Randomly from birth records, on age and sex	Use of supplements	Interviewer-administered questionnaire
Jensen et al. [11] (NCCLS ^d^)	Northern California, 1995–1999	ALL	0–14	138/138	Community controls matched on age, sex, Hispanic ethnicity and maternal race	Dietary intake, micronutrient and supplement use	Interviewer-administered questionnaire
Kwan et al. [35] (NCCLS ^d^)	California, 1995–2002	Acute leukemia	0–14	311/398	Community controls matched on age, sex, Hispanic ethnicity and maternal race	Use of supplements	Interviewer-administered questionnaire
Kwan et al. [14] (NCCLS ^d^)	California, 1995–2002	ALL	0–14	282/359	Community controls matched on age, sex, Hispanic ethnicity and maternal race	Dietary intake and micronutrients	Interviewer-administered questionnaire
Linabery et al. [36] (COG ^e^)	USA, Canada, 1996–2006	ALL, AML	<1 year	443/324	Community controls matched on birth year and location of residence	Use of supplements	Telephone interviews
McKinney et al. [32]	Scotland, 1976–1994	Acute leukemia	0–14	144/716	Community controls matched on age, sex, region (for four studies)	Use of supplements	Interviewer-administered questionnaire
Menegaux et al. [20] (ELECTRE ^a^)	France, 1995–1998	ALL, AML	0–14	534/567	Community controls matched on age, sex, region	Coffee consumption	Self-administered standardized questionnaire
Menegaux et al. [21] (ADELE ^a^)	France, 1995–1999	ALL, AML	0–14	280/288	Hospitalized controls (for diseases other than cancer or birth defects) matched on age, gender and center	Coffee and tea consumption	Interviewer-administered questionnaire
Milne et al. [43] (Aus-ALL ^b^)	Australia, 2003–2007	ALL	0–14	416/1361	Prospectively recruited by random digit dialing and matched on age, sex and residence	Use of supplements	Mailed, self-administered questionnaires and Food Frequency Questionaries
Milne et al. [23] (Aus-ALL ^b^)	Australia, 2003–2007	ALL	0–14	393/1249	Prospectively recruited by random digit dialing and matched on age, sex and residence	Coffee and tea consumption	Mailed, self-administered questionnaires and Food Frequency Questionaries
Ognjanovic et al. [44] (COG ^e^)	USA, Canada, 1996–2006	ALL, AML	<1 year	434/323	Community controls matched on birth year and location of residence	Use of supplements	Telephone interview
Orsi et al. [24] (ESTELLE ^a^)	France, 2010–2011	ALL, AML	0–14	747/1421	Community controls matched on age and gender	Coffee and tea consumption	Structured telephone questionnaires
Peters et al. [16]	California, USA, 1980–1987	Acute leukemia	0–10	232/288	Community controls matched on age, ethnicity and sex by random digit dialing	Dietary intake and cola drink consumption	Interviewer-administered questionnaire
Petridou et al. [25]	Greece, 1993–1994	Acute leukemia	0–14	153/300	Hospitalized controls matched on age and sex, contemporaneously hospitalized for minor conditions	Coffee consumption	Interviewer-administered questionnaire
Petridou et al. [13]	Greece, 1999–2003	ALL	0–4	131/131	Hospitalized controls matched on age and sex, contemporaneously hospitalized for minor conditions	Dietary intake	Interviewer-administered questionnaire
Robison et al. [33] (CCG ^f^)	USA, 1980–1984	AML	0–14	204/203	Controls were chosen with random digit dialing and matched on age, race and location	Use of supplements	Interview
Ross et al. [15] (CCG ^f^)	USA, 1983–1988	AML	<18 months	84/97	Controls were chosen with random digit dialing and matched on age, race and location	Dietary intake, coffee, tea and cola drink consumption and use of supplements	Medical records and interview
Sarasua et al. [12]	USA, 1973–1986	ALL	0–14	56/206	Community controls matched by age, sex and telephone exchange area. By random digit dialing	Dietary intake	Medical records and interview
Schuz et al. [37]	Germany, 1992–1997	ALL, AML	0–14	755/2057	Community controls matched on age, sex and residence	Use of supplements	Self-administered questionnaire and telephone interview
Shaw et al. [38]	Canada, 1980–2000	Acute leukemia	0–14	789/789	Community controls matched on age and sex	Use of supplements	Telephone interview
Singer et al. [17] (NCCLS ^d^)	California, 1995–2008	ALL, AML	0–14	784/1,076	Community controls matched on age, sex, Hispanic ethnicity and maternal race	Dietary intake	Interviewer-administered questionnaire
Singer et al. [29] (NCCLS ^d^)	California, 1995–2008	ALL, AML	0–14	784/1,076	Community controls matched on age, sex, Hispanic ethnicity and maternal race	Micronutrient intake and maternal supplement use	Interviewer-administered questionnaire
Thompson et al. [45]	Australia, 1984–1992	ALL	0–14	98/166	Controls matched for age and sex randomly selected from the state electoral roll	Use of supplements	Self-administered questionnaire, interview and medical records
Van Steensel-Moll et al. [34]	Netherlands, 1973–2010	ALL	0–14	519/507	Community controls matched on age, sex and residence	Use of supplements	Mailed questionnaires
Wen et al. [39] (CCG ^f^)	USA, Canada, Australia, 1989–1993	ALL	0–14	1842/1986	Controls were chosen with random digit dialing and matched on age, race and location	Use of supplements	Telephone interview
Pooled analysis							
Karalexi et al. [18] (CLIC ^g^)	France, Germany, Greece, USA, 1999–2003	AML	0–14	444/1255	Community controls matched on age and sex	Coffee and tea consumption	Self-administered questionnaire and telephone interview
Metayer et al. [46] (CLIC ^g^)	Australia, Canada, France, Germany, New Zealand and the United States (from CLIC) and Brazil, Costa Rica, Egypt and Greece, 1980–2012	ALL, AML	0–14	7548/11,635	Controls participating in age-matched studies	Use of supplements	Self-administered questionnaire and telephone interview
Milne et al. [19] (CLIC ^g^)	France, Australia, Greece, USA, 1995–2016	ALL	0–14	2552/4876	Community controls (matched by study variables)	Coffee and tea consumption	Food frequency questionnaires and general questionnaires

^a^ ADELE, ELECTRE, ESCALE and ESTELLE studies are part of Etude Sur les Cancers et les Leucémies de l’Enfant, Study on Environmental and Genetic Risk Factors of Childhood Cancers and Leukemia, ADELE and ELECTRE, which studied leukemias from 1995 to 1998; ESCALE (2002–2003) and ESTELLE (2010–2011) studied cancer and leukemia. ^b^ Aus-ALL: Australian Study of Causes of Acute Lymphoblastic Leukemia in Children. ^c^ UKCCS: United Kingdom Childhood Cancer Study. ^d^ NCCLS: Northern California Childhood Leukemia Study. ^e^ COG: Children’s Oncology Group. ^f^ CCG: Children’s Cancer Group. ^g^ CLIC: Childhood Leukemia International Consortium.

**Table 5 ijerph-20-05428-t005:** Overview of the association between maternal food group intake during pregnancy (if not specified) and the risk of acute leukemia in children.

Reference	Type of Leukemia	Food Group	OR	95% CI		Adjustments
Jensen et al. [11] ^&^	ALL	Vegetables	0.53	0.33–0.85 *	Total energy intake, income, history of	
		Fruits	0.71	0.49–1.04	miscarriage or stillbirth, hours of exposure to other children in day care, indoor exposure to insecticide during pregnancy and the proportion of foods reported as great or very large	
		Grain products	1.6	0.37–1.98		
		Dairy products	1.16	0.78–1.72		
		Protein sources	0.4	0.18–0.90 *		
		Cured meat	0.71	0.44–1.15		
		Fat, sweets and snacks	1.18	0.67–2.06		
Kwan et al. [14] ^&^	ALL	Vegetables (excludes salad, potatoes, soup and stew)	0.65	0.50–0.84 *	Adjusted for total energy intake, household income, indoor insecticide exposure during pregnancy and proportion of foods reported as large or extra-large portion size	
		Fruit (excludes fruit juice)	0.81	0.65–1.00 *		
		Grain products	1.2	0.70–2.05		
		Dairy products	1.06	0.83–1.35		
		Legumes	0.75	0.59–0.95 *		
		Protein sources	0.55	0.32–0.96 *		
		Cured meat	0.91	0.78–1.05		
Peters et al. [16]	Acute leukemia	≥12 servings/month:			Self-reported use of indoor pesticides, hair dryers, black-and-white televisions and fathers’ occupational exposure to spray paint during pregnancy and to other chemical exposures post-pregnancy	
		Ham, bacon and sausage	1	0.50–2.00		
		Hot dogs	2.4	0.70–8.10		
		Bologna, pastrami, corned beef and lunch meat	1.3	0.80–2.40		
		Hamburgers	1.2	0.50–2.50		
		Charbroiled meats	0.9	0.50–1.80		
		≥30 servings/month:				
		Oranges or orange juice	0.8	0.50–1.40		
		Grapefruit or grapefruit juice	1.1	0.50–2.70		
		Apple or apple juice	0.9	0.50–1.40		
Petridou et al. [13]	ALL	per quintile increase:			Sex and age, maternal age at birth, birth weight, maternal smoking during pregnancy, maternal education level, job occupation and total daily energy intake during pregnancy	
		Vegetables	0.76	0.60–0.95 *		
		Fruits	0.72	0.57–0.91 *		
		Grain products	1.23	0.94–1.60		
		Milk and dairy products	0.82	0.66–1.02		
		Meat and meat products	1.25	1.00–1.57 *		
		Butter and margarine	1.41	0.97–2.06		
		Sugar and syrup	1.32	1.05–1.67 *		
		Pulses and nuts	0.96	0.77–1.20		
		Fish and seafood	0.72	0.59–0.89 *		
Ross et al. [15]	Acute leukemia	Vegetables ^#^ (daily)	2.8	1.20–6.40 *	Adjusted for maternal education	
		Fruits ^#^ (daily)	1.6	0.70–3.30		
		Beans ^#^ (≥1/week)	1.2	0.50–3.10		
		Butter (≥1/week)	1.4	0.70–3.00		
		Cured meats (≥4/week)	1	0.50–2.20		
		Fish (≥4/week)	0.5	0.20–1.10		
Sarasua et al. [12]	ALL	≥1 serving/week:			Adjusted for other types of meat (dichotomized), age at diagnosis and per capita income	
		Ham, bacon and sausage	1.5	0.70–3.00		
		Hot dogs	0.9	0.40–1.80		
		Hamburgers	1.2	0.50–2.70		
		Charcoal-broiled foods	1	0.50–1.90		
Singer et al. [17] ^&^	ALL	1 serving/4184 kJ (1000 kcal)			Maternal Hispanic ethnicity, household income, mother’s education, father’s education, maternal age category and vitamin supplement use	
		Vegetables	0.97	0.86–1.10		
		Fruits	0.7	0.52–0.94 *		
		Dairy products	1.01	0.84–1.22		
		Dietary fiber from beans (1 g)	0.95	0.88–1.02		
		Protein (10 g)	0.91	0.79–1.05		
		Fatty acid ratio	1.07	0.78–1.45		
		Trans fat (1 g)	1.07	0.99–1.16		
		Percentage of energy content from sweets (10%)	1.09	0.94–1.26		
	AML	1 serving/4184 kJ (1000 kcal):				
		Vegetables	0.84	0.54–1.30		
		Fruits	0.23	0.08–0.70*		
		Dairy products	0.87	0.48–1.57		
		Dietary fiber from beans (1 g)	1.03	0.80–1.34		
		Protein (10 g)	1	0.63–1.59		
		Fatty acid ratio	1.08	0.42–2.77		
		Trans fat (1 g)	1.11	0.85–1.44		
		Percentage of energy content from sweets (10%)	1.4	0.84–2.34		

* = Statistically significant results. ^#^ Food that is an a priori topoisomerase II inhibitor. ^&^ These studies evaluated the intake during the year prior to the index pregnancy to represent the probable nutritional status at the beginning of the pregnancy.

**Table 6 ijerph-20-05428-t006:** Overview of the association between maternal coffee consumption during pregnancy and risk of acute leukemia in children.

Reference	Type of Leukemia	Consumption (Cups per Day) Never as Reference	OR	95% CI	Adjustments
Cohort study					
Madsen et al. [27]	Acute leukemia	0.5–3	0.89	0.48–1.65	Adjusted for cohort (Danish National Birth/Aarhus Birth), smoking, maternal age at birth and parity
		>3	1.37	0.56–3.32	
Case-control study					
Bonaventure et al. [22]	Acute leukemia	<1	1.00	0.80–1.30	Gender, age, birth order, breastfeeding, maternal education and parental socio-professional category
		1–2	1.30	1.00–1.70 *	
		>2	1.60	1.20–2.10 *	
	ALL	<1	1.30	0.70–2.10	
		1–2	1.80	1.00–3.30 *	
		>2	2.40	1.30–4.30 *	
	AML	<1	1.00	0.80–1.30	
		1–2	1.30	1.00–1.70 *	
		>2	1.50	1.10–2.00 *	
Clavel et al. [26]	Acute leukemia	<3	1.60	0.90–2.90	Age, gender, center, origin and parental socio-professional category
		≥3	4.10	1.60–10.10 *	
Menegaux et al. [21]	ALL	≤3	1.10	0.70–1.80	Child’s age, sex, center, origin
		4–8	2.40	1.30–4.70 *	
		>8	3.10	1.00–9.50 *	
	ANLL	≤3	1.60	0.60–4.30	
		4–8	2.80	0.70–10.40	
		>8	3.00	0.30–35.10	
Menegaux et al. [20]	ALL	≤3	1.10	0.80–1.40	Adjusted for age, gender, region, socio-professional category and birth order
		>3	1.40	0.90–2.40	
	AML	≤3	1.60	0.80–3.00	
		>3	1.40	0.50–4.40	
Milne et al. [23]	ALL	0–1	0.77	0.51–1.16	Age, sex, state of residence, maternal age, mother’s country of birth and parent education
		>1	1.12	0.72–1.74	
Orsi et al. [24]	Acute leukemia	≤1	0.80	0.70–1.00	Age, sex, mother’s age at child’s birth, mother’s education and birth order
		>1–2	1.00	0.70–1.30	
		>2	1.10	0.90–1.50	
	ALL	≤1	0.80	0.60–1.10	
		>1–2	1.00	0.70–1.30	
		>2	1.30	1.00–1.70 *	
	AML	≤1	0.90	0.60–1.50	
		>1–2	1.10	0.60–1.90	
		>2	0.50	0.20–1.10	
Petridou et al. [25]	Acute leukemia	>2	0.89	0.55–1.46	Gender, age and place of residence
Ross et al. [15]	Acute leukemia	≤3/week	1.50	0.70–3.30	Adjusted for maternal education
		≥4/week	2.50	1.00–6.20 *	
	ALL	≤3/week	1.10	0.40–3.00	
		≥4/week	2.30	0.70–8.20	
	AML	≤3/week	2.40	0.60–9.20	
		≥4/week	2.60	0.70–10.00	
Pooled analysis					
Karalexi et al. [18]	AML	1	1.03	0.74–1.43	Child’s age, sex, ethnicity, maternal age, household socioeconomic status, maternal smoking, birth weight
		>1	1.40	1.03–1.92 *	
		1 cup per day increment	1.18	1.01–1.39 *	
Milne et al. [19]	ALL	Any	1.04	0.94–1.19	Child’s age, sex and ethnicity, study of origin, birth order, birth year, maternal age and education, household socioeconomic status, maternal smoking during pregnancy and breastfeeding
		>0–1	0.95	0.84–1.07	
		>1–2	1.07	0.92–1.25	
		>2	1.27	1.09–1.48 *	

* = Statistically significant results.

**Table 7 ijerph-20-05428-t007:** Overview of the association between maternal tea consumption during pregnancy and risk of acute leukemia in children.

Reference	Type of Leukemia	Consumption (Cups per Day) Never as Reference	OR	95% CI	Adjustments
Case control study					
Bonaventure et al. [22]	Acute leukemia	<1	1.10	0.80–1.50	Gender, age, birth order, breastfeeding, maternal education and parental socio-professional category
		1	0.80	0.70–1.10	
		>1	0.90	0.70–1.20	
	ALL	<1	1.50	0.80–2.70	
		1	0.90	0.50–1.60	
		>1	0.50	0.20–1.10	
	AML	<1	1.10	0.80–1.50	
		1	0.80	0.60–1.10	
		>1	1.00	0.70–1.30	
Menegaux et al. [21]	ALL	≤3	1.20	0.80–1.90	Child’s age, sex, center, origin
		>3	1.20	0.60–2.60	
	ANLL	≤3	0.60	0.30–1.40	
		>3	–	–	
Milne et al. [23]	ALL	Any	0.82	0.56–1.18	Age, sex, state of residence, maternal age, mother’s country of birth and parent education
		0–1	0.81	0.54–1.22	
		>1	0.82	0.54–1.23	
Orsi et al. [24]	Acute leukemia	≤1	0.90	0.60–1.20	Age, sex, mother’s age at child’s birth, mother’s education and birth order.
		1	0.80	0.70–1.10	
		>1	1.10	0.80–1.40	
	ALL	≤1	0.90	0.60–1.20	
		1	0.80	0.60–1.10	
		>1	1.00	0.80–1.40	
	AML	≤1	0.90	0.40–1.90	
		1	1.20	0.70–2.10	
		>1	1.00	0.50.–1.80	
Ross et al. [15]	AML	≤3/week	0.50	0.10–2.30	Adjusted for maternal education
		≥4/week	0.30	0.10–1.70	
Pooled analysis					
Karalexi et al. [18]	AML	1	0.95	0.68–1.33	Child’s age, sex, ethnicity, maternal age, household socioeconomic status, maternal smoking, birth weight
		>1	0.70	0.42–1.15	
		1 cup per day increment	0.87	0.70–1.08	
Milne et al. [19]	ALL	Any	0.94	0.85–1.03	Child’s age, sex and ethnicity, study of origin, birth order, birth year, maternal age and education, household socioeconomic status, maternal smoking during pregnancy and breastfeeding
		>0–1	0.95	0.83–1.08	
		>1–2	0.93	0.76–1.15	
		>2	1.02	0.83–1.25	

**Table 8 ijerph-20-05428-t008:** Overview of the association between maternal soft drink consumption during pregnancy and risk of acute leukemia in children.

Reference	Type of Leukemia	Consumption Never as Reference	OR	95% CI	Adjustments
Bonaventure et al. [22]	Acute leukemia	1 glass/day	1.10	0.80–1.50	Gender, age, birth order, breastfeeding, maternal education and parental socio-professional category
		>1 glass/week	1.30	1.00–1.60 *	
		>1 glass/day	1.30	1.00–1.80 *	
	ALL	1 glass/day	0.80	0.40–1.80	
		>1 glass/week	1.30	0.80–2.20	
		>1 glass/day	1.10	0.50–2.10	
	AML	1 glass/day	1.20	0.80–1.60	
		>1 glass/week	1.20	1.00–1.60 *	
		>1 glass/day	1.30	1.00–1.80 *	
Peters et al. [16]	Acute leukemia	≥30 glasses/month	1.00	0.60–1.60	Adjustments for all factors thought to be potential confounders did not affect these associations
Ross et al. [15]	Acute leukemia	≤3 glasses/week	1.60	0.70–3.60	Adjusted for maternal education
		≥4 glasses/week	0.90	0.40–2.00	
	ALL	≤3 glasses/week	2.60	0.90–7.50	
		≥4 glasses/week	1.00	0.30–2.80	
	AML	≤3 glasses/week	0.70	0.10–3.00	
		≥4 glasses/week	0.60	0.10–2.30	

* = Statistically significant results.

**Table 9 ijerph-20-05428-t009:** Overview of the association between maternal micronutrient intake during pregnancy (or period specified) and risk of acute leukemia in children.

Reference	Type of Leukemia	Micronutrient Intake	OR	95% CI	Adjustments
Folate					
Bailey et al. [28]	ALL	Energy-adjusted dietary folate (mcg)			Age, sex and state of residence in Australia
		<395	1.00		
		>395 to 454	0.68	0.44–1.06	
		>454 to 524	0.58	0.37–0.91 *	
		>524 to 624	0.44	0.27–0.71 *	
		>624	0.70	0.44–1.12	
		Energy-adjusted dietary B6 (mg)			
		<1.39	1.00		
		>1.39 to 1.54	1.04	0.67–1.62	
		>1.54 to 1.67	1.15	0.74–1.81	
		>1.67 to 1.85	1.28	0.82–2.00	
		> 1.85	1.60	1.02–2.51 *	
		Energy-adjusted dietary B12 (mcg)			
		<3.18	1.00		
		>3.18 to 3.75	0.72	0.47–1.10	
		>3.75 to 4.27	0.79	0.52–1.21	
		>4.27 to 5.34	0.85	0.56–1.31	
		>5.34	0.49	0.31–0.77 *	
Jensen et al. [11] ^&^	ALL	Non-users of vitamin supplements (continuous)			Energy intake, income, miscarriages or stillbirths, exposure to other children at preschools, indoor exposure to insecticide during pregnancy and the proportions of the food
		Folate (dietary folate equivalents)	–	–	
		Vitamin B6 (mg)	–	–	
		Vitamin B12 (mcg)	–	–	
		Alpha carotene (mcg)	0.66	0.42–1.05	
		Total glutathione (mg)	0.15	0.02–0.96 *	
		Reduced glutathione (mg)	0.19	0.03–1.07	
Kwan et al. [14] ^&^	ALL	Median daily intake (continuous)			Energy intake, household income, indoor insecticide exposure during pregnancy and proportion of foods reported as large or extra-large portion size
		Folate (dietary folate equivalents)	1.02	0.71–1.47	
		Vitamin B6 (mg)	1.12	0.73–1.72	
		Vitamin B12 (mcg)	1.10	0.82–1.48	
		Provitamin A carotenoids (mcg)	0.77	0.60–0.98 *	
		Alpha-carotene (mcg)	0.78	0.65–0.93 *	
		Total glutathione (mg)	0.48	0.25–0.90 *	
		Reduced glutathione (mg)	0.49	0.27–0.90 *	
Singer et al. [29] ^&^	ALL	From food only (continuous, one unit change)			Adjusted for parents’ education, household income, maternal age at child’s birth and energy intake. Models for all mothers also adjusted for mother’s ethnicity. Models for nutrient intake from food only also adjusted for intake of B
		Folate (100 dietary folate equivalent/day)	0.99	0.93–1.06	
		Vitamin B6 (1 mg/day)	0.91	0.74–1.12	
		Vitamin B12 (1 mcg/day)	0.97	0.92–1.02	
	AML	From food only (continuous, one unit change)			
		Folate (100 dietary folate equivalent/day)	0.90	0.76–1.07	
		Vitamin B6 (1 mg/day)	0.47	0.23–0.98 *	
		Vitamin B12 (1 mcg/day)	0.86	0.73–1.02	

* = Statistically significant result. ^&^ These studies evaluated the intake from the year before to the index pregnancy to represent the probable nutritional status at the beginning of pregnancy.

**Table 10 ijerph-20-05428-t010:** Overview of the association between maternal supplement use during pregnancy (or period specified) and the risk of acute leukemia in children.

Reference	Type of Leukemia	Type of Supplementation (vs. No-Use in the Index Period)	OR	95% CI	Adjustments
Case control study					
Abudaowd et al. [31]	Acute leukemia	Folate	0.181	0.076–0.43 *	Univariate analysis
Ajrouche et al. [40]	Acute leukemia	Folic acid (before)	0.70	0.50–1.00 *	Age, gender, birth order, maternal educational and maternal age at child’s birth
		Folic acid	0.80	0.50–1.20	
	ALL	Folic acid (before)	0.70	0.50–1.10	
		Folic acid (first trimester)	1.10	0.90–1.50	
		Folic acid (second trimester)	1.10	0.90–1.50	
		Folic acid (third trimester)	1.20	0.90–1.60	
	AML	Folic acid (three months before)	0.40	0.10–1.20	
		Folic acid (first trimester)	1.00	0.50–1.70	
		Folic acid (second trimester)	1.10	0.60–2.00	
		Folic acid (third trimester)	1.20	0.70–2.20	
Amigou et al. [41]	Acute leukemia	Folic acid (pre-conception or first trimester)	0.30	0.20–0.60 *	Age, gender and socioeconomic status
		Folic acid (second trimester)	0.60	0.30–1.00 *	
		Folic acid (third trimester)	0.30	0.10–0.70 *	
		Folic acid	0.40	0.30–0.60 *	
		Folic acid or multivitamin	0.60	0.50–0.80	
	ALL	Folic acid (pre-conception or first trimester)	0.30	0.20–0.70 *	
		Folic acid (second trimester)	0.60	0.30–1.10	
		Folic acid (third trimester)	0.20	0.10–0.80 *	
		Folic acid	0.40	0.30–0.60 *	
		Folic acid or multivitamin	0.70	0.50–0.90 *	
	ANLL	Folic acid (pre-conception or first trimester)	0.20	0.00–1.40	
		Folic acid (second trimester)	0.50	0.10–2.00	
		Folic acid (third trimester)	0.40	0.10–2.80	
		Folic acid	0.30	0.10–0.90 *	
		Folic acid or multivitamin	0.60	0.30–1.10	
Bailey et al. [28]	ALL	Any folate supplement use	1.05	0.75–1.48	Age, sex and state of residence
		Any B6 or B12 supplement use	1.22	0.87–1.71	
Bonaventure et al. [30]	ALL	Iron	1.36	1.14–1.63 *	Year of birth
	AML		1.26	0.86–1.85	
Dockerty et al. [42]	ALL	Folic acid (with or without iron)	1.10	0.50–2.70	Age, sex, marital status and mother’s education
		Multivitamins	0.80	0.20–3.10	
		Other vitamin or mineral supplements	1.50	0.70–3.10	
		Iron (with or without folic acid)	1.20	0.70–2.10	
		Iron without folic acid	1.30	0.80–2.30	
Jensen et al. [11] ^&^	ALL	Average daily intake, continuous (before pregnancy)			Total energy intake, income, history of miscarriage or stillbirth, hours of exposure to other children in day care, indoor exposure to insecticide during pregnancy and the proportions of food as great or very large
		Vitamin A	0.58	0.32–0.98 *	
		Folic acid	0.78	0.33–1.81	
		Vitamin B6	0.84	0.41–1.75	
		Vitamin B12	0.86	0.45–1.45	
		Iron	0.89	0.51–1.53	
Kwan et al. [35]	Acute leukemia	Iron (3 months before pregnancy and during breastfeeding)	0.70	0.51–0.97 *	Household income, maternal education, maternal age at birth
		Iron (3 months before and during pregnancy)	0.76	0.52–1.11	
	ALL	Iron (3 months before pregnancy and during breastfeeding)	0.67	0.47–0.94 *	
		Iron (3 months before and during pregnancy)	0.72	0.47–1.09	
Kwan et al. [14] ^&^	ALL	Median daily intake, continuous (before pregnancy)			Adjusted for total energy intake, household income, indoor insecticide exposure during pregnancy and proportion of foods reported as large or extra-large portion size
		Vitamin A	0.82	0.62–1.08	
		Folate	1.02	0.71–1.47	
		Vitamin B6	1.12	0.73–1.72	
		Vitamin B12	1.10	0.82–1.48	
		Iron	1.05	0.77–1.44	
Linabery et al. [36]	ALL	Any prenatal vitamins	0.63	0.34–1.18	Race/ethnicity, household income
		Vitamins (periconceptional)	0.77	0.54–1.11	
		Vitamins	0.66	0.39–1.11	
		Vitamins (the year before and during pregnancy)	0.77	0.55–1.09	
		Iron (before pregnancy)	1.22	0.82–1.80	
		Iron (periconceptional)	1.30	0.62–2.72	
		Iron	1.22	0.81–1.84	
		Iron (the year before and during pregnancy)	3.04	0.77–12.03	
	AML	Any prenatal vitamins	1.20	0.53–2.73	Race/ethnicity, household income
		Vitamins (periconceptional)	1.05	0.68–1.61	
		Vitamins	1.05	0.55–2.04	
		Vitamins (the year before and during pregnancy)	0.88	0.60–1.31	
		Iron (before pregnancy)	0.82	0.51–1.33	
		Iron (periconceptional)	1.26	0.55–2.88	
		Iron	0.77	0.46–1.27	
		Iron (the year before and during pregnancy)	1.72	0.33–9.06	
McKinney et al. [32]	Acute leukemia	Iron supplements	0.95	0.62–1.45	Maternal age
	ALL	Iron supplements	0.80	0.51–1.25	
Milne et al. [43]	ALL	Folate (first trimester)	1.19	0.91–1.56	Age, sex, residence, ethnicity, education, birth order and maternal age
		Folate (second and third trimester)	0.83	0.65–1.06	
		Iron (first trimester)	1.08	0.43–2.75	
		Iron (second and third trimester)	1.04	0.57–1.89	
Ognjanovic et al. [44]	ALL	Multivitamin	0.71	0.51–1.00 *	Child’s year of birth
	AML		0.79	0.53–1.17	
Robison et al. [33]	ALL	Vitamins/Iron	1.00	0.51–1.96	Adjusted for potential confounding factors
Schuz et al. [37]	ALL	Vitamins, folate and/or iron	0.84	0.69–1.01	Sex, age, year of birth, degree of urbanization and socioeconomic status
	AML		1.13	0.74–1.72	
Shaw et al. [38]	ALL	Vitamins with folic acid	1.00	0.80–1.20	Maternal education and maternal age at birth
		Other vitamins	1.00	0.70–1.30	
Singer et al. [29] ^&^	ALL	Continuous, unite change (before pregnancy)			Father’s and mother’s education, household income, maternal age at child’s birth and energy intake. Models for all mothers also adjusted for mother’s ethnicity. Models for nutrient intake from food only also adjusted for intake of B vitamin-containing supplements (yes/no)
		Folate (100 dietary folate equivalent/day)	0.97	0.94–1.01	
		Vitamin B12 (1 mg/day)	0.96	0.93–1.00 *	
		Vitamin B6 (1 mg/day)	0.89	0.79–1.00 *	
	AML	Continuous, unite change (before pregnancy)			
		Folate 100 dietary folate equivalent/day	0.93	0.85–1.03	
		Vitamin B6 1 mg/day	0.72	0.51–1.04	
		Vitamin B12 1 mg/day	0.92	0.84–1.02	
Thompson et al. [45]	ALL	Folate with or without iron	0.40	0.21–0.73 *	Univariate analysis
		Iron or folate	0.37	0.21–0.65 *	
		Iron and folate	0.41	0.22–0.75 *	
		Iron alone	0.75	0.37–1.51	
Van Steensel-Moll et al. [34]	Acute leukemia	Iron preparations	1.30	0.90–2.00	Age and sex
Wen et al. [39]	ALL	Vitamins	0.70	0.50–1.00 *^†^	Immuno-phenotype, sex, age, household income, maternal and paternal race, education, smoking and drinking before or during pregnancy
		Iron supplements	0.90	0.70–1.00 *	
Pooled analysis					
Metayer et al. [46]	ALL	Folic acid (any time)	0.80	0.71–0.89 *	Age, sex, ethnicity, parental education and study
		Folic acid (pre-conception)	0.82	0.70–0.96 *	
		Folic acid	0.77	0.67–0.88 *	
		Folic acid (before and during pregnancy)	0.78	0.78–0.91 *	
		Vitamins (any time)	0.85	0.78–0.92 *	
		Vitamins (pre-conception)	0.82	0.79–0.99 *	
		Vitamins	0.81	0.74–0.88 *	
		Vitamins (before and during pregnancy)	0.78	0.69–0.88 *	
	AML	Folic acid (any time)	0.68	0.48–0.96 *	
		Folic acid (pre-conception)	0.88	0.59–1.32	
		Folic acid	0.52	0.31–0.89 *	
		Vitamins (any time)	0.92	0.75–1.14	
		Vitamins (pre-conception)	0.96	0.66–1.39	
		Vitamins	0.85	0.64–1.14	

* = Statistically significant results; ^†^ = 99% CI; ^&^ These studies evaluated the intake during the year before the index pregnancy to represent the probable nutritional status at the beginning of pregnancy.

**Table 11 ijerph-20-05428-t011:** Summary Odds Ratios (ORs) and 95% Confidence Intervals (CIs) obtained in meta-analysis of data from referenced published studies of association of childhood leukemia with maternal dietary intake.

	Leukemia Type	No. of Studies	References	OR (95% CI)	Heterogeneity I2, *p*
Food group					
Fruits	Acute leukemia	–		–	–
	ALL	2	[13,17]	0.71 (0.59–0.86)	0%, 0.0003
	AML	–		–	–
Vegetables	Acute leukemia	–		–	–
	ALL	2	[13,17]	0.88 (0.69–1.11)	69%, 0.28
	AML	–		–	–
Grains	Acute leukemia	–		–	–
	ALL	2	[13,17]	1.22 (0.96–1.56)	0%, 0.10
	AML	–		–	–
Dairy products	Acute leukemia	–		–	–
	ALL	2	[13,17]	0.92 (0.75–1.12)	51%, 0.41
	AML	–		–	–
Processed meat (cured meat)	Acute leukemia	–		–	–
	ALL	2	[13,17]	1.01 (0.68–1.51)	37%, 0.96
	AML	–		–	–
Non-alcoholic drinks					
Coffee	Acute leukemia	5	[22,24,25,26,27]	1.36 (0.97–1.91)	68%, 0.08
	ALL	5	[20,21,22,23,24]	1.45 (1.12–1.89)	35%, 0.005
	AML	4	[20,21,22,24]	1.18 (0.66–2.13)	45%, 0.56
Tea	Acute leukemia	2	[22,24]	0.97 (0.80–1.51)	0%, 0.78
	ALL	4	[22,23,24]	0.95 (0.79–1.14)	0%, 0.56
	AML	2	[22,24]	1.00 (0.73–1.37)	0%, 1.00
Soft drinks	Acute leukemia	2	[16,22]	1.23 (0.97–1.55)	0%, 0.08
	ALL	–		–	–
	AML	–		–	–
Supplements					
Folic Acid	Acute leukemia	2	[40,41]	0.55 (0.28–1.09)	84%, 0.08
	ALL	6	[28,29,40,41,43,45]	0.77 (0.59–1.01)	89%, 0.06
	AML	3	[29,40,41]	0.87 (0.56–1.36)	59%, 0.54
Iron	Acute leukemia	3	[32,34,35]	0.94 (0.65–1.38)	68%, 0.30
	ALL	7	[30,32,35,39,42,43,44]	0.96 (0.76–1.21)	73%, 0.35
	AML	–		–	–

For each included factor, quantitative analysis was performed with at least two studies; results from studies restricted to infants, univariate analysis with no adjustments and overlapping data were excluded.

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
