# Peer review of "Role of Maternal Diet in the Risk of Childhood Acute Leukemia: A Systematic Review and Meta-Analysis"

_ijerph, 2023, doi:10.3390/ijerph20075428_

Round 1

Reviewer 1 Report

The manuscript investigates the role of maternal diet in the risk of childhood acute leukemia: a 2 systematic review and meta-analysis. The manuscript can be accepted for publication after minor corrections.

 -L18 Change ‘leukemia (the most common type of cancer in children)’ to’ leukemia, one of the most common types of cancer in children,’

L32-33: suggest removing ‘systematic…937’

L47: change ‘teens’ to ‘adolescents’.  To standardize the term used as per L40.

Suggest removing ‘….’ In Table 1

Table 5b Please arrange the data in chronological order based on year published

Author Response

Dear reviewer (1):

Thank you for your observations and comments. We have addressed them:

 -L18 Change ‘leukemia (the most common type of cancer in children)’ to’ leukemia, one of the most common types of cancer in children,’

We accepted your suggestion, and we modified the text accordingly.

Many studies have investigated the etiology of acute leukemia, one of the most common types of cancer in children, however, there is a lack of clarity regarding preventable risk factors.

L32-33: suggest removing ‘systematic…937’

We accepted your suggestion, and we modified the text accordingly.

prevention strategies applicable at the population level. Review Registration: PROSPERO registration no. CRD42019128937.

L47: change ‘teens’ to ‘adolescents’.  To standardize the term used as per L40.

We accepted your suggestion, and we modified the text accordingly.

Although acute leukemia is the most common cancer in children and adolescents, it is a rare disease; therefore, studying its etiology remains challenging.

Suggest removing ‘….’ In Table 1

We accepted your suggestion, and we modified the text accordingly.

Studies without original data (editorial, review, reports, guidelines)

Table 5b Please arrange the data in chronological order based on year published.

In this regard, the alphabetical order based upon first author’s last name was chosen for tables 4 and 5a-f over chronological order based upon year published (chronological order based upon year was applied if there was more than one publication form the same author). This is because we consider it easier for the reader to locate the authors rather than year of publication, considering that the text also refers to the author which is more unique than the year.

Reviewer 2 Report

Dear Authors,

Thank you for the opportunity to review the original article entitled „Role of maternal diet in the risk of childhood acute leukemia: a systematic review and meta-analysis

WHAT IS THE MAIN QUESTION ADDRESSED BY THE RESEARCH? The topic of this article is to summarize the current evidence on the potential effect of maternal diet on the risk of acute leukemia in children. The results of this study are important because they offer consistent arguments in favor of potential dietary risk factors.

IS THE TOPIC ORIGINAL OR RELEVANT IN THE FIELD? The topic of the study is relevant in the field of nutrition and cancer because of the limited knowledge of the role of diet and nutrients on the risk factors for acute leukemia in children and adolescents.

WHAT SPECIFIC IMPROVEMENTS SHOULD THE AUTHORS CONSIDER REGARDING THE METHODOLOGY?

The study is correctly designed and technically sound. The methods used in this research are well described and provide sufficient details to be understood. The research methodology is in line with the proposed objectives.

However, I have two observations: In the exclusion criteria were mentioned studies from a study in 1 specific village, and children from 1 hospital. Some hospitals could be regional centers for treating children with leukemia. You should be more specific regarding the reasons for rejecting these studies from the analysis. The limitation section of the article should be extended, to discuss the results from the studies which assess the presence of carcinogens in the maternal diet that for methodological reasons could not be included in the analysis.

Author Response

Dear reviewer (2):

Thank you for your observations and comments. We addressed them accordingly:

In the exclusion criteria were mentioned studies from a study in 1 specific village, and children from 1 hospital. Some hospitals could be regional centers for treating children with leukemia. You should be more specific regarding the reasons for rejecting these studies from the analysis.

We thank the reviewer for this comment. Indeed, we were careful while choosing publications, and this was not the main criterion to exclude studies. There were few publications performed in one hospital / centre, but they were excluded only after applying the quality control by Fowkes et al. criteria. To avoid confusion, we removed this criterion “studies from a study in 1 specific village, and children from 1 hospital” from the exclusion criteria as this was not the main criterion for exclusion (Table 1).

Small studies (e.g. less than 50 cases)

The limitation section of the article should be extended, to discuss the results from the studies which assess the presence of carcinogens in the maternal diet that for methodological reasons could not be included in the analysis.

Thank you for this observation, indeed by design this systematic review is focused on the results from studies that evaluated maternal diet intake and the risk of childhood acute leukemia, and not on biomarker focused studies or potential mechanisms explaining these relationships. The limitation section was rephrased, and we nuanced the discussion about the potential mechanisms.

However, there are new generation studies (biomarker based, which are not the focus of this review) that can support some of these findings. Studies have been conducted to assess the presence of carcinogens (such as acrylamide, nitrosamines, and others) in the maternal diet, which can be found in the placenta, linked to fetal hemoglobin, and can also be found in newborn lymphocytes. Since these biomarkers were also found in healthy children, further studies focusing on genetic variants found some polymorphisms that could affect the expression of enzymes such as folate hydrolase 1. With lower detoxifying activity, DNA damage or higher activity of carcinogens may occur [64].

Reviewer 3 Report

Leukemia is one of the most frequently diagnosed cancers in children and it is not yet known which factors could be increasing the risk, including factors related to eating habits. It is a very good review.

The following comments are sent to the authors for their consideration:

1. In the last paragraph of the introduction it is mentioned that there are factors of the maternal diet before, during and after pregnancy on the development of leukemia. In this study, what types of studies were included? That is, were studies included where diet was measured before, during and after pregnancy, or only at some point in time.

2. How were the food and drink groups selected to analyze their risk for leukemia? It is necessary to place this information in methodology. Why choose coffee, tea, sweet drinks specifically?

3. It would be important for table 4 on the characteristics of the included studies to add information on the moment in which each study collected the information (before, during or after pregnancy).

4. In the review there is a study where the consumption of vegetables has a greater risk of developing leukemia. How can you explain it? Could it be the effect of insecticides?

5. Why was the analysis stratified by age groups not carried out?

6. The authors conclude that the consumption of cured meats has a risk of developing leukemia, however, none of the results of this study show it.

7. It is suggested that the authors discuss the memory bias when the diet information was collected in relation to the results obtained.

8. Authors are suggested to consider reducing the number of tables, that is, if associations are not found in many studies, the effect could only be reported in the category of leukemia, but not by type of leukemia.

9. It would be necessary to discuss in the study about the heterogeneity between the studies.

10. How is this review different from other reviews and meta-analyses?

Author Response

Dear reviewer (3):

Thank you for your observations and comments. We addressed them accordingly:

  1. In the last paragraph of the introduction it is mentioned that there are factors of the maternal diet before, during and after pregnancy on the development of leukemia. In this study, what types of studies were included? That is, were studies included where diet was measured before, during and after pregnancy, or only at some point in time.

Thank you for highlighting this. The current review indeed focuses on maternal diet, but originally included other peri-pregnancy and early infancy risk factors. We deleted the misleading part of the sentence so that it fully reflects the scope of the current review.

It provides an overview of existing data linking the influence of maternal diet, supplementation, and consumption of non-alcoholic beverages on the development of acute leukemia in children.

  1. How were the food and drink groups selected to analyze their risk for leukemia? It is necessary to place this information in methodology. Why choose coffee, tea, sweet drinks specifically?

As explained in the methods section, the search was done via a stepwise approach, with first a broad search using general terms like maternal exposure, diet, and nutrition. In the 2nd search we included specific terms based upon the outcome of the first search to make sure we covered all relevant literature for each exposure found during the 1st search.

  1. It would be important for table 4 on the characteristics of the included studies to add information on the moment in which each study collected the information (before, during or after pregnancy).

Only the cohort study included in this review collected the information before pregnancy, the other studies (case control studies) collected the information after pregnancy.

  1. In the review there is a study where the consumption of vegetables has a greater risk of developing leukemia. How can you explain it? Could it be the effect of insecticides?

Ross et al. 1996, investigated specific food items, because of their content of DNA topoisomerase II inhibitors, and the development of Acute Myeloid Leukemia in infants. This research had a very specific hypothesis. We now underlined this better in the review to improve the interpretation of this unexpected result.

  1. Why was the analysis stratified by age groups not carried out?

Of all the studies included, only one had information about adolescents (Abudaowd et al. 2021), and it was not added in the meta-analysis due to the lack of adjustments.

  1. The authors conclude that the consumption of cured meats has a risk of developing leukemia, however, none of the results of this study show it.

Thank you for your observation, some studies focused on cured meats, we stated that there are no statistically significant results. Now, we have removed the reference to cured meats from the conclusion section.

This systematic review of the available literature indicates that a maternal diet rich in vegetables, fruits, and supplementation with folic acid and multivitamins during pregnancy could be inversely associated with the risk of childhood AL, while a higher intake of coffee, and/or caffeinated beverages could increase the risk.

  1. It is suggested that the authors discuss the memory bias when the diet information was collected in relation to the results obtained.

Thank you for your observation, we consider this aspect as a limitation (included in the discussion section).

  1. Authors are suggested to consider reducing the number of tables, that is, if associations are not found in many studies, the effect could only be reported in the category of leukemia, but not by type of leukemia.

We are aware of the number and extension of tables, but for transparency we decide to keep all the results and not only the significative ones. It is also important to keep the results as they were conducted, and consider that ALL, and AML have potential different pathways to develop. The summary table 6, not only summarizes the results, but the empty spaces highlight the lack of studies to be included.

  1. It would be necessary to discuss in the study about the heterogeneity between the studies.

Thank you for pointing this out. We added a comment about this in the strengths and limitations section.

We also noticed that some studies restricted their search to specific food items [18] and given the heterogeneity of how results were reported (above all the studies), making difficult to compare results.

  1. How is this review different from other reviews and meta-analyses?

Before conducting our systematic review, some previous systematic reviews had been conducted. However, these systematic reviews only focused on specific aspects, for example: Dessypris 2017 focused only on maternal diet (food groups and nutrients) and Metayer 2014 focused on the use of supplements. There were others systematic review that focused on coffee consumption (Milne 2018 and Karalexi 2019). In this review, we added new information from new studies published after those previous systematic reviews. We also consider to give a more comprehensive review, highlighting some pregnancy factors that could have an impact on primary prevention. In addition, we would like to increase the interest on this important, though often neglected topic.

Round 2

Reviewer 3 Report

The authors responded appropriately to the comments made.